# Whole and Purified Aqueous Extracts of *Nigella sativa* L. Seeds Attenuate Apoptosis and the Overproduction of Reactive Oxygen Species Triggered by p53 Over-Expression in the Yeast *Saccharomyces cerevisiae*

**DOI:** 10.3390/cells11050869

**Published:** 2022-03-03

**Authors:** Wafa Mihoubi, Emna Sahli, Fatma Rezgui, Najeh Dabebi, Rabiaa Sayehi, Hajer Hassairi, Najla Masmoudi-Fourati, Kamel Walha, Khalifa ben Khadhra, Mohamed Baklouti, Imen Ghzaiel, Sami Fattouch, Hela Menif, Raja Mokdad-Gargouri, Gérard Lizard, Ali Gargouri

**Affiliations:** 1Laboratoire de Biotechnologie Moléculaire des Eucaryotes, Centre de Biotechnologie de Sfax, Université de Sfax, B.P 1177, Sfax 3018, Tunisia; najehdabebi5@gmail.com (N.D.); sayehirabiaa23@gmail.com (R.S.); raja.gargouri@cbs.rnrt.tn (R.M.-G.); 2Laboratoire d’Analyses, Centre de Biotechnologie de Sfax, Université de Sfax, B.P 1177, Sfax 3018, Tunisia; emna.sahli@yahoo.fr (E.S.); fatmarezgui1983@gmail.com (F.R.); hassairihajer@hotmail.fr (H.H.); masmoudinajla2004@yahoo.fr (N.M.-F.); walha_kamel@yahoo.fr (K.W.); yahyakhalifa33@yahoo.com (K.b.K.); 3Institut Supérieur de Biotechnologie de Sfax, Université de Sfax, Sfax 3018, Tunisia; moha.baklouti@gmail.com; 4Lab-NAFS ‘Nutrition—Functional Food & Vascular Health’, Faculty of Medicine, University of Monastir, LR12ES05, Monastir 5000, Tunisia; imenghzaiel93@gmail.com; 5Laboratoire de Biochimie du Peroxysome, Inflammation et Métabolisme Lipidique, Université de Bourgogne/Inserm, 21000 Dijon, France; gerard.lizard@u-bourgogne.fr; 6Laboratoire d’Ecochimie, INSAT, University of Carthage, Tunis 1073, Tunisia; sami.fattouch@insat.rnu.tn; 7Laboratoire d’Hématologie, Faculté de Médecine de Sfax, Sfax 3027, Tunisia; helamenif@yahoo.fr

**Keywords:** apoptosis, black cumin, *Nigella sativa* L. aqueous extract, oxidative stress, p53, *Saccharomyces cerevisiae*, yeast

## Abstract

Plants are an important source of pharmacologically active compounds. In the present work, we characterize the impact of black cumin (*Nigella sativa* L.) aqueous extracts on a yeast model of p53-dependent apoptosis. To this end, the Saccharomyces cerevisiae recombinant strain over-expressing p53 was used. The over-expression of p53 triggers the expression of apoptotic markers: the externalization of phosphatidylserine, mitochondrial defect associated with cytochrome-c release and the induction of DNA strand breaks. These different effects were attenuated by *Nigella sativa* L. aqueous extracts, whereas these extracts have no effect on the level of p53 expression. Thus, we focus on the anti-apoptotic molecules present in the aqueous extract of *Nigella sativa* L. These extracts were purified and characterized by complementary chromatographic methods. Specific fluorescent probes were used to determine the effect of the extracts on yeast apoptosis. Yeast cells over-expressing p53 decrease in relative size and have lower mitochondrial content. The decrease in cell size was proportional to the decrease in mitochondrial content and of mitochondrial membrane potential (ΔΨm). These effects were prevented by the purified aqueous fraction obtained by fractionation with different columns, named C4 fraction. Yeast cell death was also characterized by reactive oxygen species (ROS) overproduction. In the presence of the C4 fraction, ROS overproduction was strongly reduced. We also noted that the C4 fraction promotes the cell growth of control yeast cells, which do not express p53, supporting the fact that this purified extract acts on cellular mediators activating cell proliferation independently of p53. Altogether, our data obtained on yeast cells over-expressing p53 demonstrate that anti-apoptotic molecules targeting p53-induced apoptosis associated with mitochondrial dysfunction and ROS overproduction are present in the aqueous extracts of Nigella seeds and in the purified aqueous C4 fraction.

## 1. Introduction

Apoptosis is an evolutionarily conserved cell-death mechanism. One of its roles is to selectively eliminate damaged cells with irreparable DNA damage. Apoptosis is characterized by typical biochemical and morphological features, such as nuclear fragmentation and chromatin condensation, cell blebbing, the externalization of phosphatidyl-serine, and caspase cascade activation, leading to inter-nucleosomal DNA fragmentation [1]. Apoptosis misregulation can result in major human diseases, such as age-related diseases, including neurodegenerative, cardiovascular and eye diseases, viral infections, and cancers [2].

It is well admitted that wild-type p53 is a tumor suppressor gene, which is implicated in DNA repair, the cell-cycle process, aging and the death of cells that have been exposed to cellular stresses induced by physical, chemical or biological agents [3,4]. The ability of p53 to activate apoptosis through different pathways may define its tumor suppressor activity to prevent tumor development [5].

In mammals, apoptosis occurs in two ways: extrinsic and intrinsic pathways. The first passes through specific pro-apoptotic receptors (cell-death receptors) present on the surface of the cell membrane, such as the tumor necrosis factor receptors (TNFRs) super-family [6,7], and the intrinsic pathway is centered on the mitochondria [8,9]. One of the most widely studied events associated with the intrinsic pathway triggered by physical and chemical agents is the MOMP (mitochondrial outer membrane permeabilization), which is required for the release of cytochrome-c from the mitochondria and the formation of the apoptosome leading to caspase cascade activation. Cytochrome-c has an essential role in ATP generation by navigating between complexes III and IV at the electron transport chain. Once outside the mitochondria, cytochrome-c binds to APAF-1 (Apoptotic Protease Activating Factor-1) that oligomerizes and forms the apoptosome complex, which is responsible of caspase activation. In addition to cytochrome-c, mitochondria release other proteins, such as Omi (called HtrA2) and SMAC (also called Diablo), which elicit caspase activity and contribute to cell death independent of the activation of caspases [10,11]. This alternate caspase-independent mode of cell death relates to the wide implication of MOMP, regardless of the mode of cell death adopted, manifesting as a loss of ΔΨm and a progressive demise of mitochondrial function [12]. A potential role of p53 in the control of mitochondrial status cannot be excluded [13] and this point remains important to be elucidated. In addition, there are also several evidences that p53 plays crucial roles in senescence and aging [14].

To date, several engineered yeast models have become a valuable aid in studying the roles of endogenous or heterologous proteins, such as p53, in order to decipher the molecular basis of complex diseases, such as degenerative disorders and cancers [15]. Yeast models can be considered as powerful approaches in the discovery of novel therapeutic agents against these pathologies [15]. Thus, p53 over-expression inhibits the growth of both budding and fission yeasts [16,17]. Previous data obtained in our laboratory also showed that the over-expression of p53 in *Saccharomyces cerevisiae* resulted in cell death with key hallmarks of apoptosis, including nuclear DNA fragmentation, phosphatidylserine externalization and ROS accumulation [18]. The cell-death pathway induced by p53 expression is not clearly understood and, in some cases, this phenotype is reported as an external stress condition [19]. Instead, we must recall that some methods validate the apoptotic cell death in yeast. Current evidence supports the idea that the “PS externalization is a universal feature of yeast cells undergoing apoptosis” as reported by by Carmona-Gutierrez et al. [19]. Similar to mammalian cells, the cell death in yeast is connected to mitochondrial processes, such as the release of cytochrome-c and other dysfunctions in *S. cerevisiae* yeast expressing p53 [20]. Other works have also shown that oxidative stress is endowed with a very important role in apoptosis, especially when death is induced by oxide derivatives of cholesterol (oxysterols), which are involved in several aged-related diseases [21]. In ROS accumulation-mediated apoptosis, chemical compounds and natural compounds, and a mixture of natural compounds, such as oils, including Nigella seed oils, have been shown to inhibit or attenuate apoptosis [22]. *Nigella sativa* L. (Ranunculaceae) seeds, better known as “black cumin” or sometimes “black seed”, have been extensively studied for their wide biological activity spectrum, especially their antioxidant properties. We already used *Nigella sativa* L. extracts as a means of decreasing cell death caused by p53 over-expression in yeast, and FTIR spectroscopy was employed to characterize apoptosis associated with p53 over-expression [23].

In the present study, we extend our investigation of apoptosis features due to p53 over-expression in the yeast system, and we study the activity of aqueous *Nigella sativa* L. extract in this model. The morphological and functional markers (cell size, mitochondrial status, and ROS production) that are induced during the p53-mediated cell death, as well as the mitochondrial content, were analyzed in the presence of aqueous *Nigella sativa* L. extract and in the control.

## 2. Materials and Methods

### 2.1. Strain and Media

The yeast strain used in our experiences was W303-1B (MATα leu2-31,12; ade2-1; trp1-1; his3-11,15 ura3-1; can1-100) [24] and is named W303 in the present study; it does not express p53. This strain was transformed by the multicopy YepDP8-1 vector named pDP, containing the Ura3 marker, the origin of replication 2 µ and the galactose-inducible promoter Gal10/Cyc1 [25]; this control strain is named W303/pDP (W303/pDP does not express p53). The wt-p53 cDNA was inserted in pDP behind the Gal10/Cyc1 promoter, as previously described [26], in the same W303-1B host strain, and is thus named W303/p53 (W303/p53 expresses p53).

The composition of the minimal medium (MM) used for yeast clones was the following: yeast nitrogen base 0.67% (Difco, Illkirch, France), adenine (120 mg/L), tryptophan (40 mg/L) and 20 mg/L of histidine and leucine. The MMGlu contained 2% glucose as a non-inducing condition, while 2% galactose was added to launch the induction of the p53 gene (MMGal). The growth kinetics was measured on solid media (containing 2% agar) or in liquid, as explained hereafter.

### 2.2. Cell-Growth Kinetics

The recombinant yeasts were cultivated in liquid MMGlu, then ten-fold serial dilutions from the cell culture were prepared, from which 5 µL was spotted onto the solid media of MMGlu and MMGal, and on the same media onto which 100 μL of Nigella extract was spread. The plates were then incubated for 2 to 3 days at 30 °C. For liquid cultures, recombinant yeast clones were grown in MMGlu overnight until they reached the stationary phase, then they were used to inoculate either the MMGlu or MMGal media at 5.10^5^ cells/mL and cultured at 30 °C and 150 rpm. The yeast growth was evaluated following the optical density (OD) measurement at 600 nm every 2 h.

### 2.3. Extraction of Proteins and Western Blot

Yeast cells were grown until they reached the stationary phase on MMGlu, then harvested by centrifugation at 1000× *g*, washed with sterile water and suspended in MMGal to be cultured for 24 h in the presence and absence of aqueous extracts of *Nigella sativa* L. Cells were collected, and their proteins and RNA content were analyzed by Western blots and RT-PCR. One mL of each culture was regularly withdrawn and the cells were harvested by centrifugation. The pellets were suspended in 100 µL of lysis buffer (150 mM NaCl, 5 mM EDTA pH 8, 50 mM Tris-HCl pH 8, 1 mM dithiothreitol) in ice and equal quantities of glass beads (size 0.5 mm) were added, then vortexed for 7–9 cycles of 30 s. The lysate’s proteins were electrophoresed on 10% (SDS-PAGE) and transferred onto a PVDF (polyvinylidene difluoride) membrane (Amersham-Biosciences, Buckinghamshire, United Kingdom). Immunological detection was performed using DO-1, a monoclonal anti-p53 antibody (Santa Cruz Biotechnology, Dallas, TX, USA) diluted at 1/1000, then, by an anti-mouse antibody, conjugated to horseradish peroxidase (BioRad, Hercules, CA, USA) and revealed by chemo-luminescence using the ECL Plus kit (Amersham-Biosciences) [18,19].

### 2.4. Nigella sativa L. Extract Preparation

A total of 1 kg of *Nigella sativa* L. seeds was washed and air dried, then it was crushed under a Kinematica Gmbh electronic device (Luzern Switzerland), which allowed the separation of the two parts of the seeds: the seed coat and the albumen. Each crushed part was mixed with distilled water at the rate of 10 g of powder/30 mL [27]. The mixture was homogenized in a polytron apparatus (Kinematica Gmbh, Eschbach, Germany), then centrifuged for 15 min at 11,200× *g*. The brownish-orange supernatant was filtered through 0.45 µm filters and stored at 4 °C until use.

### 2.5. Preparative High-Performance Liquid Chromatography (HPLC)

The 0.45 μM filtered plant extract was injected on a PL Aquagel OH 40–10 µm (300 × 25 mm) column of a preparative HPLC (Knaeur system, model 1100) composed of a wellchem K-1800 preparative pump and an eurochrom K-2501 UV detector. A 5 mL/min flow rate was applied, and the peaks were detected at 280 nm, according to an isocratic phase composed of water. The collected fractions were concentrated into a rotavator apparatus.

### 2.6. Preparative Fast-Performance Liquid Chromatography (FPLC)

The 0.45 μm filtered active fraction was deposited on a preparative FPLC anionic unoQ12 column (Biorad composed of model 2128 collector). The flow rate was 3 mL/min and UV monitoring was performed at 280 nm. The elution of fractions was performed using a linear NaCl gradient as in [22]. Indeed, the chromatography program constituted of 1–3 min of isocratic buffer 100% A (tris-citrate pH 5), two sample injections, 3–20 min of constant flow with 100% A, followed by an upward gradient of 100% B-buffer (0.1 M NaCl in A-buffer), 4–5 min of constant flow of 100% B-buffer, and 5 min of a constant flow of 100% A-buffer. It should be noted that tris-acetate pH 5 buffer produced a better separation of peaks than tris-HCl pH 8.

### 2.7. Analytical High-Performance Liquid Chromatography (HPLC)

Filtered active Q1 fraction was deposited on a C18 column (C18 reversed-phase column (TC 01107 Eurospher 100-5 C18′′ 250 × 8 mm)) of an HPLC Agilent System 1100 (Santa Clara, CA, USA). A flow rate of 0.5 mL/min was applied to elute the fractions and UV monitoring was performed at 280 nm.

The linear elution gradient consisted of solvents A and B (1% formic acid and 100% methanol, respectively), starting with 95% A and 5% B at 0 mins, reaching 75% A after 10 min, followed by a regular drop of 5% A every 5 min until it reached 20% A after 60 min, followed by a constant plateau for 10 min at 95% A before the following injection. The flow rate applied was of 0.5 mL/min.

Active C4 fraction was deposited on the hydrophilic Biorad column (Aminex HPX-87-H (300 × 7.8 mm) of an HPLC-DAD Agilent 1260 infinity (Marnes-la-Coquette, France), using an isocratic mobile phase composed of 0.01% H_2_SO_4_, monitored using a UV and RID detector.

### 2.8. Precipitation with Cold Acetone/Ethanol

Aqueous *Nigella sativa* L. extract (N-ext) was precipitated with cold acetone or an ethanol sample in the proportions (v-Next/v-Solvent), (*v*/2*v*), and (*v*/3*v*) at −20 °C for 24 h, then centrifuged at 16,000× *g* for 20 min. The pellet was dissolved in the same v mL of water and the supernatant was dried and suspended in v mL of water.

### 2.9. Radical Scavenging Activity

The radical scavenging assay was conducted as described by Mansouri et al. [28]. The DPPH solution was prepared using 0.25 mg DPPH (2,2-diphenyl-1-picryl-hydrazyl-hydrate) dissolved in 10 mL of methanol. An aliquot (25 µL) of extract or the antioxidant standard (BHT: butylated hydroxy toluene) was added to 975 µL of the DPPH solution. The mixture was vigorously shaken and incubated for 30 min at room temperature in the dark. The decrease in the absorbance value was measured at 517 nm. The percentage of DPPH scavenging activity was calculated using the following equation:

% DPPH scavenging activity = 100 × (Abs of control − Abs of sample/Abs of control), where the Abs of the sample is the absorbance of the reaction with the test compounds, and the Abs of the control is the absorbance of the control reaction mixture without the test compounds.

The IC50 value is the concentration of the extract causing 50% neutralization of the DPPH radicals, which was calculated from the plot of the inhibition percentages against the concentration.

### 2.10. Survival Assay

The W303/p53 and W303/pDP cells (previously grown on MMGlu until reaching the stationary phase) were diluted in sterile water and spread onto MMGlu, MMGal and “MMGal supplemented by extract or purified molecule”, and incubated for 3 days at 30 °C. The number of colonies was determined.

The absolute cell count (total number of cells) was determined by flow cytometry. An alternative to the CFU count for cell viability assessment is the determination of cell viability by flow cytometry after cell staining with propidium iodide (PI, 5 µg/mL); the cells were grown in MMGal with and without the C4 fraction. The mean fluorescence intensity of PI fluorescence was used to estimate cell death (an increase in the mean fluorescence intensity of PI comparatively to the control cells indicates an increase in the cell death and consequently a loss of cell viability). A cyflow^®^Space flow cytometer was used. Data analyses were performed using Winlist software (Verity software). A total of 10,000 event data was collected using a logarithmic amplification. All assays were run in duplicate. Data analyses were performed by the use of Winlist software (Verity software).

### 2.11. Annexin V and Propidium Iodide (PI) Assay: Flow Cytometric Assay

The externalization of phosphatidylserine (PS) was assessed by Annexin-V/PI assay, as described by Madeo et al. (1997) [29], with slight modifications. After being centrifuged, the yeast cells were resuspended in 1.2 M Sorbitol, 100 mM KH_2_PO_4_ pH 7.4. Spheroblasts were obtained by incubation in a lysis buffer (1.2 M Sorbitol, 100 mM KH_2_PO_4_, pH 7.4, 0.2% β-mercaptoethanol), and 0.2 mg/mL of zymolyase for 1 h at 37 °C. Spheroblasts were washed and suspended in 198 µL of binding buffer (10 mM Hepes buffer pH 7.4, 140 mM NaCl, 2.5 mM CaCl2, 1.2 M sorbitol) and incubated with 2 µL of Annexin V per 105 cells for 15 min at room temperature. Stained spheroblasts with Annexin V were washed with binding buffer, 2 µL of PI was added and the samples were immediately analyzed by flow cytometry on an Attune^®^ Acoustic Focusing Cytometer (Thermo Fisher Scientific, Dardilly, France) equipped with two lasers (red and blue); Annexin V was measured at 494/518 nm (FL1-A) and PI was measured at 535/617 nm (FL2-A). The data were collected from 10,000 yeast cells using logarithmic amplification. Each assay was run in duplicate and, as a control, unstained cells were used. Data analysis was investigated using Winlist software (Verity software, Topsham, ME, USA).

### 2.12. Determination of Relative Cell Size and Granularity by Flow Cytometry

The cell size was simultaneously determined by the forward scatter channel (FSC) and side scatter channel (SSC), using the flow cytometer cyflow^®^Space (Sysmex-Partec, Münster, Germany) equipped with a 488 argon laser. Fluorescent 10 µm beads were used to calibrate the flow cytometer in relation to size and fluorescence. The voltage for the forward scatter channel (FSC) was fixed at 120, and for the side scatter channel (SSC) at 200. The flow rate was adjusted to 120 µL/min. The data of 10,000 events were collected using logarithmic amplification. All the assays were run in duplicate. Data analyzes were performed using Winlist software (Verity software).

### 2.13. Flow Cytometric Quantification of the Relative Mitochondrial Content

The mitochondrial content was determined using the Mito-Tracker Green FM fluorescent dye (Thermo Fisher Scientific). Yeast cells were cultured in MMGal under continuous agitation for 24 h at 30 °C. The cells were harvested by centrifugation, washed twice in PBS and resuspended in PBS in order to obtain 10^7^ cells/mL per sample. MitoTracker Green was added at the final concentrations of 100 and 500 nM [30], and the mixture was incubated at room temperature for 30 min. Then, the cells were immediately monitored by flow cytometry. The amount of fluorescence intensity correlates with the mitochondrial quantity. The single-cell fluorescence was measured using a 488 argon laser-equipped with a cyflow^®^Space flow cytometer (Sysmex-Partec, Münster). Mitotracker Green FM fluorescence was measured using the green 536 ± 25 nm band-pass filter (FL1). Fluorescent 10 µm beads were used to calibrate the flow cytometer in fluorescence. The flow rate was adjusted to 120 µL/min. a total of 10,000 events data were collected using logarithmic amplification. All the assays were run in duplicate and the unstained cells were used as controls. Data analyses were performed using Winlist software (Verity software).

### 2.14. Assessment of Intracellular Reactive Oxygen Species (ROS) Levels by Flow Cytometric Analysis Using H_2_-DCFDA

The cells were cultured in MMGal under continuous agitation at 30 °C for 24 h. Protoplasts were obtained, as it was described for Annexin V staining, washed twice in PBS and resuspended in PBS. The ROS was detected by incubation for 2 h with either 10 µg/mL [31] or 24 µg/mL [32] of dichloro-dihydro-fluorescein diacetate H_2_-DCFDA. This dye is deacylated to dichloro-dihydro-fluorescein H_2_-DCF and oxidized by ROS to fluorescent dichloro-fluorescein (DCF). Then, the cells were immediately monitored by flow cytometry. The same cyflow^®^Space flow cytometer was used to measure the single cell fluorescence. The green 536 ± 25 nm band-pass filter (FL1) was used to measure the DCF fluorescence. The flow rate was adjusted to 120 µL/min and collected using logarithmic amplification. All the assays were run in duplicate and the unstained cells were used as a control. Data analyses were performed using Winlist software (Verity software). A total of 10,000 event data were regularly collected using logarithmic amplification. All the assays were run in duplicate and the unstained cells were used as controls. Data analyses were performed by the use of Winlist software (Verity software).

### 2.15. Flow Cytometric Assessment of Mitochondrial Membrane Potential after Staining with DiOC_6_(3)

Cells were grown in MMGal in the presence and absence of the Nigella fraction, and were harvested, double washed and resuspended in PBS at 10^7^ cells/mL per sample, then stained by 100 and 40 nM of DiOC_6_(3). The mean fluorescence intensity of the fluorescence was used to measure the mitochondrial membrane potential (ΔΨm) using the same cyflow^®^Space flow cytometer and the same parameters as in Section 2.14.

### 2.16. Flow Cytometric Assessment of Mitochondrial Membrane Potential after Staining with JC-1

The mitochondrial membrane potential (∆Ψm) was also measured by staining cells with JC-1 (5, 50, 6, 60 -tetrachloro-1, 10, 3, 30 -tetraethylbenzimidazolocarbo-cyanine iodide) (Life Technology). The cells were cultured in MMGal under continuous agitation at 30 °C for 24 h. Protoplasts were obtained, as it was described for Annexin V staining. Protoplasts were washed twice in PBS and re-suspended in PBS then incubated with JC-1 (0.25 µmoles/106 cells/mL) for 60 min at 35 °C. After washing in PBS 1X, the samples were analyzed in the cytometer A cyflow^®^Space flow cytometer (Sysmex-Partec). JC-1 selectively enters mitochondria as a monomer and emits green fluorescence (FL1), or it can form aggregates in the case of polarized/energized mitochondria, and the fluorescence is collected on an FL2 filter (orange fluorescence). The fluorescence of the JC-1 monomers was measured using the green 536 ± 25 nm band-pass filter (FL1), while the fluorescence of JC-1 aggregates was measured using the orange 590 ± 25 nm band-pass filter (FL2). The flow rate was adjusted to 120 µL/min. The data of 10,000 events were collected using a logarithmic amplification. All the assays were run in duplicate. Data analyses were performed by the use of the Winlist software (Verity software).

### 2.17. Statistical Analysis

The results are expressed as the mean ± standard deviation (SD). Student′s *t*-test was used to assess the statistical significance of the differences found; *p* < 0.05 is considered as statistically significant, and the significant differences are represented by an asterisk (*****).

## 3. Results

### 3.1. Anti-Apoptotic Activity of Aqueous Nigella sativa L. Extract

The convenient p53 yeast genetic model [18] resulted in our consideration of the screening for molecules that might antagonize the negative effect of p53 on yeast viability. In this model, under repressing conditions (MMGlu), the W303/p53 and W303/pDP yeast cells were able to grow, while, upon the induction conditions (MMGal), the p53 protein inhibited the growth of the recombined cells [18] (Figure 1A,B). As it is known that in such conditions, the p53 expressing yeasts accumulate reactive oxygen species (ROS) [18].thus the *Nigella sativa* L. seeds were chosen as the raw materials, for their richness in several types of antioxidant compounds [33]. Figure 1A shows that the spreading of Nigella aqueous extracts on the galactose medium permitted the growth of p53 recombinant yeast cells.

In order to verify these results, the yeast cell growth was monitored on liquid MMGal, either supplemented or not by the aqueous Nigella extracts. As already reported [18], on MMGal without Nigella extract, the growth of W303/p53 was very slow (Figure 1B). In the presence of Nigella extracts, the p53 yeast cells were able to grow appropriately (Figure 1B). Interestingly, the growth of the W303/pDP control clone was also improved.

### 3.2. Effect of Aqueous Nigella Extract on p53 Expression

Western blot analysis was carried out on the total proteins extracted from W303/p53 recombinant yeast cells cultured for 24 h in MMGal, in the presence or absence of aqueous extracts of Nigella. As it is shown in Figure 1C, the presence of the P53 protein on the migration path of the proteins extracted from W303/p53 grown on MMGal in the presence of Nigella, clearly proves that that the Nigella extract did not affect p53 protein expression. These data show that aqueous Nigella extracts did not affect the expression level of p53.

### 3.3. Purification of Anti-Apoptotic Molecules from Aqueous Nigella sativa L. Extract

Grinding Nigella seeds separated the two parts of *Nigella sativa* L. seeds: the integument (or seed coat) and the albumen. An aqueous extract from each part was prepared and tested for the protective effect against apoptosis. The active molecules were found in the tegument and this was the first step of purification. Bioassay-guided fractionation was applied in order to purify and characterize the anti-apoptotic molecules. The extract was passed on an “aquagel OH-40” column to be fractionated by size. A single major active fraction (G2) was eluted at 20.90 min (Figure 2A). When heated at 100 °C, both the G2fraction and the whole crude extract did not lose their anti-apoptotic powers. Partition chromatography of the G2 fraction and Nigella extract using ethyl acetate, chloroform and hexane revealed that the active molecules are contained in the aqueous phase. The resistance to chloroform weakens the hypothesis that the active molecule is proteinaceous.

In the next step, G2 was passed on a cationic FPLC column. The active molecules were not retained by the column, at pH 5, being concentrated in the Q1 fraction (Figure 2B). Nevertheless, this unoQ column was efficient in discarding several non-active molecules, (Figure 2B). Various effectors (EDTA, beta mercaptoethanol and SDS) added at concentrations of 5, 10 and 20 mM to the Q1 fraction, as well as its treatment by proteinase K, did not affect its activity. The results obtained once more support the fact that the anti-apoptotic molecules are not of proteinaceous nature.

Thereafter, an eluted fraction Q1 was passed through an RP-C18 reversed-phase column (Figure 2C). Only one fraction, P4, eluted at 27 min with 80% methanol, presented the anti-apoptotic activity. Several inactive fractions were also eluted (Figure 2C) and discarded. P4 was finally loaded on an Aminex HPX-87H polar column using a UV detector (Figure 2D). One active fraction (C4) was detected at 260 nm, at 21.82 min, while minor inactive fractions were visible on the chromatogram. The assessment of the purification yield showed that the active molecule represents 77 mg per 1 kg of Nigella seeds. Efforts are in progress to purify a larger quantity of the active compound in order to complete its identification by physical and chemical analyses.

### 3.4. Effect of the C4 Purified Fraction on Cell Viability

Cell viability was assessed by the percentage of “Colony Forming Units”. This percentage was 85% in W303/pDP and 7% in W303/p53. In presence of aqueous *Nigella sativa* L. extracts (Figure 3) plated on MMGal, the cell viability of the recombinant yeast W303/p53 increased to 65.5% and that of W303/pDP was also improved, and increasing to 95%.

A similar effect was shown in the liquid cultures of both strains in the presence of the C4 fraction, by cell counting and PI staining using flow cytometry (Appendix A).

### 3.5. Antioxidant Activity of the C4 Fraction from Aqueous Nigella sativa L. Extract: An Evaluation by the DPPH Assay

As a result of the convenience and ease of spectrophotometric DPPH reduction, it has now become widely used in the evaluation of free-radical scavenging activity in vitro [28]. The antioxidant potential of the C4 fraction was estimated by evaluating the DPPH reduction at 517 nm. The IC50 values for DPPH scavenging activities of the *Nigella sativa *L. C4 fraction and the reference BHT were, respectively, 6.2 ± 0.2 and 0.46 ± 0.05 mg/mL. The concentration required to reduce the initial DPPH concentration by 50% (EC50) that is determined by the relation EC50 = IC50/DPPH (mg/mL) was 2.58 ± 0.05 and 0.19 ± 0.02, respectively.

### 3.6. Effect of the the C4 Fraction of Aqueous Nigella sativa L. Extract on the Apoptosis Markers

Figure 4 depicts the flow cytometry analysis of phosphatidylserine (PS) externalization by the FITC-Annexin/propidium iodide (PI) co-staining of W303/p53 and W303/pDP yeast cells grown in MMGal supplemented or not by the C4 fraction of aqueous *Nigella sativa *L. extract. No apoptotic cells were detected in the W303/pDP clone on MMGal, without and with the C4 fraction (Figure 4). Apoptotic cells were detected in W303/p53 (37.8%) (Figure 4). It is worth noting that, in the presence of C4, there is a remarkable decrease in the percentage of apoptotic cells (5.7%) (Figure 4). Since PS externalization is a criteria of reference for apoptosis study, the corresponding result clearly demonstrates the anti-apoptotic activity of the C4 fraction of the aqueous *Nigella sativa* L. extract.

### 3.7. Effect of the C4 Fraction of Aqueous Nigella sativa L. Extract on Cell Size

Cell death is associated with the modifications of cell size that can be measured by light scattered by flow cytometry. The distribution profile of the relative yeast cell size in the different recombinant yeast strains was evaluated with the forward scatter (FSC) values determined by flow cytometry. In the absence of the C4 fraction, the decrease in the FSC of W303/p53 indicated a decrease in size that could be explained by the morphological alterations occurring during apoptosis (Figure 5). Interestingly, the C4 fraction prevents the decrease in FSC, and similar FSC values are observed in W303/pDP, W303/pDP + C4 and W303/p53 + C4 (Figure 5). The cell size of the control strain, W303/pDP, was also slightly increased when the yeast cells were grown in MMGal + *Nigella sativa* L. C4 (Figure 5).

### 3.8. Evaluation of the Mitochondrial Content

During apoptosis, cell alterations are usually associated with several mitochondrial changes. To study the changes in the mitochondrial content, MitoTracker Green FM was used. This dye accumulates in the matrix of the mitochondria and binds to the mitochondrial proteins [34]. Thus, the “mean fluorescence intensity” (MFI) of the dye is proportional to the mass of the mitochondrial inner membrane and allows for the evaluation of the mitochondrial content [35]. The MFI signal of the MitoTracker Green fluorescence representing the mitochondria content per cell is shown in Figure 6A. The mitochondrial content was lower in W303/pDP than in W303/p53 (Figure 6B). In the presence of the C4 fraction, the content of mitochondria was increased in both strains. Thus, in the presence of C4, the mitochondrial mass was slightly increased in W303/pDP and strongly increased in W303/p53 (Figure 6).

### 3.9. Effect of the C4 Fraction of Aqueous Nigella sativa L. Extract on the Reactive Oxygen Species (ROS) Production

Reactive oxygen species are considered to be essential for the regulation of normal physiological functions, such as proliferation and cell-cycle progression, migration, differentiation, and cell death [36]. However, elevated amounts of intracellular ROS can induce cell death [18]. As the C4 fraction was able to prevent apoptosis, we asked whether this effect was due to antioxidant activities. Therefore, we determined the ROS production to evaluate the antioxidant power of the C4 fraction. The assessment of the ROS levels was measured with H_2_-DCFDA that presents the fluorescent DCF in the presence of ROS (Figure 7A). In the different assays, the intracellular ROS level was calculated from the difference of the fluorescence of the stained cells minus the fluorescence of the unstained cells (Figure 7B). Interestingly, higher levels of ROS were observed in W303/p53 than in W303/pDP, supporting a potential relationship between ROS overproduction and apoptosis. In addition, our data support the antioxidant properties of the C4 fraction: the ROS level strongly decreased in W303/p53 (Figure 7).

### 3.10. Effect of the C4 Fraction of Aqueous Nigella sativa L. Extract on the Mitochondrial Membrane Potential (ΔΨm)

As the C4 fraction had an effect on ROS hyper-production, we further wondered if it was able to act on ΔΨm. This suggests a relationship between ROS overproduction, the induction of apoptosis and the ΔΨm value during the apoptotic process associated with p53 over-expression in yeast. To this end, ΔΨm was measured by staining cells with DiOC_6_(3), a lipophilic dye. In this condition, the cells with low ΔΨm failed to accumulate DiOC_6_(3) [36,37]. In the different assays, the ΔΨm was calculated from the difference of the fluorescence of the stained cells minus the fluorescence of the unstained cells (Figure 1A). As shown in Figure 8A,B, W303/p53 exhibited a lower fluorescence intensity, compared to W303/pDP, meaning that the ΔΨm of apoptotic W303/p53 yeast cells was dissipated. In the presence of the C4 fraction, DiOC_6_(3) fluorescence intensity was increased, even in W303/pDP. This result shows that the C4 fraction increases the ΔΨm. However, in the presence of the C4 fraction, the ΔΨm value remains lower than in W303/pDP (Figure 8A,B).

For more investigations, and because of the technical limitation of DiOC_6_ (3) that stains mitochondria at low concentrations, but also stains the nuclear envelope and ER at higher concentrations [37], and because yeast cells can accumulate different amounts of DiOC_6_(3) according to the stage of growth [37], JC-1, which is more reliable to assess the ΔΨm potential [38], was used. This probe changes in fluorescence from green to orange, according to the mitochondrial value (cells with polarized mitochondria are green; cells with depolarized mitochondria are orange/red). Indeed, in healthy cells, the probe aggregates (designed as J-aggregates) inside the mitochondria having a high membrane potential, and produces an orange/red fluorescence when excited with a blue light [39]. In apoptotic cells exhibiting a drop of ΔΨm, this dye remains in a monomeric form and produces a cytoplasmic green fluorescence when excited with a blue light [40,41]. As shown in Figure 9A,B, few J-aggregates are formed in the W303/p53 spheroblasts, compared to W303/pDP, which indicates that W303/p53 is endowed with a lower mitochondrial membrane potential. In the presence of the *Nigella sativa* L. fraction, the red fluorescence of J-aggregates evidently increased in both stains. Furthermore, the red-to-green ratio of JC-1 can also be considered for the direct assessment of mitochondrial polarization. Despite of the limitation that there is a large quantity of monomeric forms in cell cytoplasms, this ratio is indeed weak in the W303p53 strain; however, it is elevated in the W303pDP strain and in W303p53 in the presence of the C4 fraction (Figure 9C). Nevertheless, J-aggregates that form at precise locations are still the ideal tool for investigating the changes in the mitochondrial membrane potential ∆Ψm. In cells with a low ∆Ψm, JC-1 exists in its monomeric form, and only shows green fluorescence. Cells with a high ∆Ψm, form J-aggregates inside the mitochondria, and thus show a red fluorescence. In the cells with a low ∆Ψm, JC-1 exists in its monomeric form, and only shows green fluorescence located in the cytoplasm and mitochondrial matrix in the depolarized mitochondria.

This result confirms that the mitochondrial membrane potential is lower in W303/p53, and that the *Nigella sativa* L. fraction prevents such modifications.

## 4. Discussion

The expression of human gene p53 in yeasts, results in cell death characterized as cell apoptosis (exposure of phosphatidylserine, DNA stand cleavage, mitochondrial defect associated with cytochrome-c release, and ROS hyper-production), and is associated with the overproduction of reactive oxygen species (ROS) [18,19]. This model is used to purify anti-apoptotic natural compounds, as it was previously shown by FTIR spectroscopy [23]. This study attempts to reconcile the previous results showing major markers by FTIR spectroscopy on P53 recombinant yeast cells, and those performed here using flow cytometry, and to display the protective role of *Nigella sativa* L. extract on yeast cells, which is studied for the first time, not as it usually reported in the literature, as pro-apoptotic [42]. Phenotypically, the cell viability of recombinant yeast p53 cultured on MMGal was significantly reduced. By adding the aqueous *Nigella sativa* L. extract, the cell viability of p53+ was restored to a certain extent, without affecting the expression of the p53 gene. We also noticed a remarkable improvement of the control of cell growth. Interestingly, the extracts also exerted a positive effect on the control p53 strain. Then, the major marker of apoptosis, PS externalization, was analyzed and showed that the purified C4 fraction from *Nigella sativa* L. extract can be qualified as anti-apoptotic. This result seems to corroborate the membrane disorder detected by FTIR spectroscopy; we also previously demonstrated that the *Nigella sativa* L. fraction prevented DNA cleavage [23].

Similar to mammalian cells, the flow cytometric analysis of the side scatter allows for a discrimination of the yeast cells committed to apoptosis that display a reduction in the cell size. This feature was associated with the mitochondrial defect [43]. Indeed, our data showed that the yeast cells with a higher mitochondrial membrane potential and mitochondrial content are those of a large size. Conversely, for yeasts that express p53 (undergoing apoptosis), the lower mitochondrial content was concomitant with the loss of cell size. Interestingly, in the presence of the *Nigella sativa* L. purified fraction, these changes are reversed/restored. Indeed, this is explained by the fact that cellular protein, and also organelle content, typically scales linearly with cell size in all types of cells, [44,45], and it is commonly expected that its functionality increases linearly with the total protein and organelle content [46,47,48]. Such a relationship occurred in a quantitative analysis of the mitochondrial network size in yeast, which revealed that mitochondrial size correlates with the cell size [48,49]. Additionally, cell size alteration and the decrease in the mitochondrial content can be explained by the mitochondrial alteration that occurred during apoptosis [50].

Our results correlate well with the key events in apoptosis that occur, particularly in the mitochondria, characterized by the cytochrome-c release [20] and the dissipation of the mitochondrial membrane potential. Several cases of Δψm losses were reported in yeast cell apoptosis. This was the case for the deletion of the gene coding for a histone chaperone, ASF1/CIA1, causing a decrease in the mitochondrial membrane potential, the dysfunction of the mitochondrial proton pump and the release of cytochrome-c into the cytoplasm [51]; this phenotype largely resembles mammalian cells. The decrease in the mitochondrial membrane potential was also observed in the yeast cell death caused by arsenite [52], edelfosine [53] and aspirin [54]. Therefore, the relation between mitochondrial dysfunction and the yeast PCD pathway is similar to the mammalian intrinsic pathway [55,56].

Thus, mitochondrial alteration results in cell size alteration and a decrease in the mitochondrial content [50].

It is known that ROS are produced during respiration in aerobic organisms or by chemical and enzymatic reactions, depending on the mitochondrial function [57], and could also be generated wherever electron transport chains exists, including the nuclear membrane and endoplasmic reticulum [58]. However, it may also be involved in the induction and regulation of the yeast apoptotic pathway [30,59]). Our group has already shown that ROS constitute the main key event in apoptosis mediated by p53 over-expression [18,59].

Elevated ROS observed in yeast over-expressing p53 suggest mitochondrial involvement. In several works, the ethanol effect on *S. cerevisiae* resulted in an increase in the intracellular ROS levels that was correlated with a marked decrease in the mitochondrial membrane potential, Δψm [60]. Papiliocin-induced yeast apoptosis, characterized by ROS accumulation, was correlated with the mitochondrial membrane potential dissipation [61]. Yeast apoptosis induced by acetic acid was also characterized by the translocation of cytochrome-c to the cytosol and elevated ROS production, both events being associated with a reduction in the consumption of oxygen and in mitochondrial membrane potential [62].

Inversely, some works on mammalian cell apoptosis reported an increase in Δψm, after a lethal stimulus, followed by a Δψm decrease in the death cascade [63]. Ludovico et al. showed that yeast apoptosis generated by acetic acid is characterized by an increase in ROS production and a transient slight hyper-polarization that was followed by depolarization, and mitochondria membrane integrity was still preserved with a lower Δψm [62]. Similarly, in *S. cerevisiae* exposed to hydrochloric acid, cell death was induced by the rise in the intracellular ROS level that was correlated with the mitochondrial hyper-polarization followed by depolarization [64]. Similar to pheromones [65], cell treated with amiodarone strongly enhanced the ROS level in yeasts, correlating with an increase in Δψm [66].

In mammalian cells, three mitochondrial responses to BAX induced a transient mitochondrial Δψm hyper-polarization, a subsequent significant depolarization of mitochondrial Δψm, and, in selected settings, the cytochrome-c release as signals of cell death [67,68]. Subsequently, it was shown that the expression of Bax in *S. cerevisiae* induces the hyper-polarization of the mitochondrial membrane, increased ROS production, growth arrest and cell death; however, under these conditions, cytochrome-c release from the mitochondria was not detected [69]. In fact, depending on the cell system, mitochondrial dysfunction may be the early key event in the apoptosis process, but it can also be the consequence of the signaling pathway of apoptosis, and may not be required for cytochrome-c release [70]. In HeLa cells, it has been shown that in the apoptosis induced by p53, there is no cytochrome-c release or ROS generation, but a transient increase in Δψ that was followed by its decrease [71].

In our case, we have already shown that in cell death induced by p53 over-expression with characteristic markers of mammalian apoptosis, ROS production [18] was associated with the mitochondrial localization of p53 and the cytochrome-c release of in the cytosol [20]. These events are now correlated with the loss of mitochondrial membrane potential and decrease in the mitochondrial content. Thus, p53 over-expression in S. cerevisiae triggers apoptosis via the mitochondrial pathway. The addition of the *Nigella sativa* L. molecule increased the survival of W303/p53, without affecting the p53 gene expression, suggesting that the anti-apoptotic molecules must affect cell physiology in a way that also affects the effects induced by p53, probably by scavenging/reducing the ROS in a direct or indirect manner, and by leading to the hyper-polarization of the mitochondrial membrane, probably by scavenging/reducing the ROS in a direct or indirect manner, and by leading to the hyper-polarization of the mitochondrial membrane. The analysis of the cell death parameters demonstrates that the *Nigella sativa* L. fraction engendered an increase in the cell size and mitochondrial content, a reduction in the intracellular ROS levels and an increase in the Δψm. These effects could result from the antioxidant power of the *Nigella sativa* L. molecule. In the literature, *Nigella sativa* L. seeds’ aqueous extract has been mentioned to have an antioxidant activity through reducing nitric oxide production by murine macrophages in vitro [27]. Many studies reported that antioxidants, such as the well-known N-acetyl-cysteine (NAC), is able to decrease the cellular mortality, to lower the oxidative free radicals produced during acid stress and to partially prevent the decrease in Δψm [64] and the fragmentation of mitochondrial filaments [51]. In this context, our group has shown that NAC can reduce the apoptosis mediated by p53 [51]. Here, we show that the *Nigella sativa* L. molecule is also endowed with an antioxidant capacity.

Interestingly enough, the addition of the *Nigella sativa* L. fraction engendered the same effects on the control strain (W303/pDP), namely the increase in the cell size and cell content of mitochondria, the reduction of ROS and the increase in Δψm. This would mean that its action is independent of the presence of p53.

Finally, we shall note that the increase in the cell size and the mitochondrial content are coupled with the protein content and respiration [46]. Mitochondrial membrane potential constitutes a vital component of mitochondrial respiration and is related to a host of mitochondrial functions, including ATP synthesis, Ca2+ 2+ homeostasis, the import of mitochondrial proteins, and metabolite transportation [72,73]. This latter behavior was already reported during the screening of positive modulators of the mitochondrial respiratory function. In this context, some molecules were shown to increase Δψm and the concentration of adenosine triphosphate (ATP), making them suitable to be applied as therapeutic tools in degenerative diseases.

## 5. Conclusions

As it is currently well-admitted that ROS overproduction, which can result from a breakdown of RedOx homeostasis in various diseases and in the aging process, can favor cell senescence and cell death, there is a great interest to identify cytoprotective molecules in this context. In addition, it is also known that the cumulative formation of ROS progressively harms mitochondrial structure and activity, leading to several cell dysfunctions and ultimately to cell death. The yeast cell model over-expressing p53 constitutes a powerful model, which allows the identification of a new properties of the biological activities of aqueous *Nigella sativa* L. seed extracts and of the corresponding purified fractions. It is worth noting that, in the yeast cells over-expressing p53, the cytoprotective activities of these aqueous extracts are characterized by an inhibition of apoptosis and of ROS overproduction. Altogether, our data support the fact that active cytoprotective molecules are present in the C4 aqueous fraction of *Nigella sativa* L. seed extracts. It is therefore expected that the characterization of the molecules present in the C4 fraction permit the development of novel drugs for the treatment of mitochondrial diseases and age-related diseases, for which few effective drugs are currently available.

## Figures and Tables

**Figure 1 cells-11-00869-f001:**
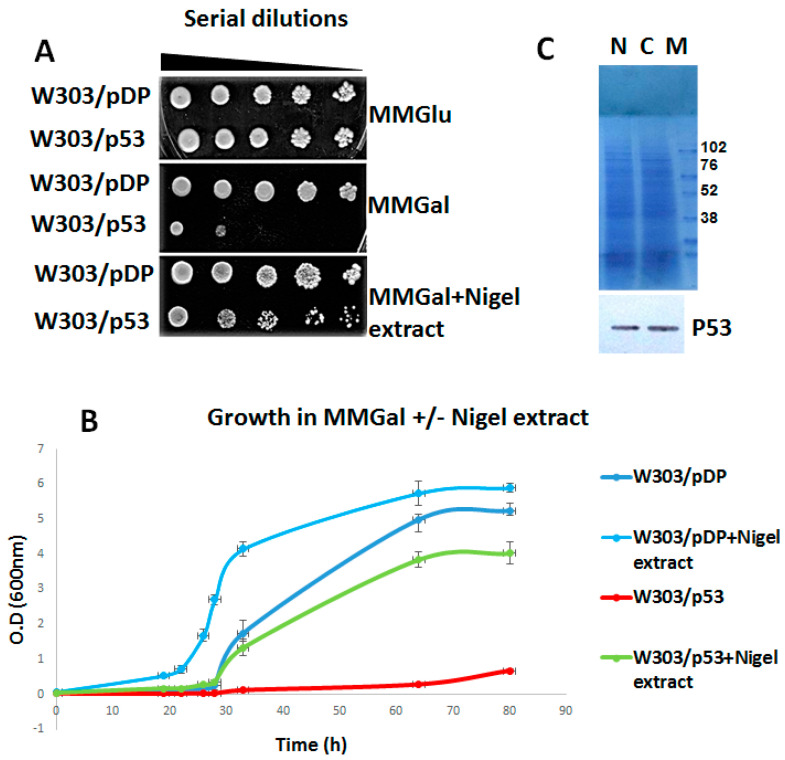
Monitoring of yeast growth, and p53 expression profile in the presence of an aqueous extract of *Nigella sativa* L.: (**A**) growth on a solid medium of serial dilution of the recombinant yeast clones W303/pDP, W303/wt on MMGlu (control), MMGal (inducible condition for p53 expression) and MMGal + Nigella extract. (**B**) Assessment of the growth of W303/pDP and W303/p53 on liquid MMgal with and without Nigella extract; cultures were performed under agitation at 30 °C and the 600 nm absorbance was measured every 2 h. (**C**) Measurement of p53 expression level in recombinant yeast W303/p53 in MMGal, with and without aqueous Nigella extract. Proteins were analyzed on a 10% SDS-PAGE and quantified by Coomassie Brilliant Blue staining, then followed by Western blot analysis with a p53-specific monoclonal antibody. N: cells cultured with aqueous Nigella extract; E: cells cultured without extract; and M: molecular weight markers.

**Figure 2 cells-11-00869-f002:**
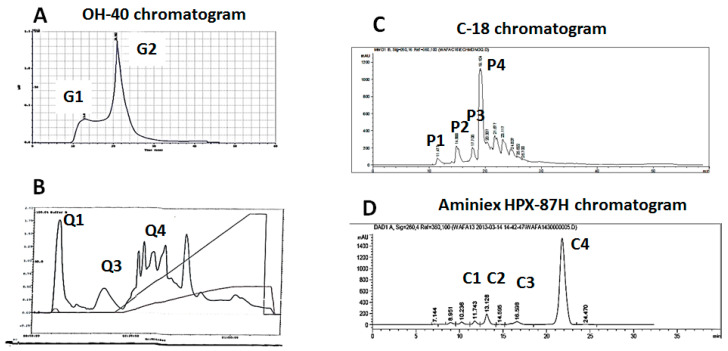
Purification by the chromatographic separations of *Nigella sativa* L. extract, leading to the identification of the active fraction C4: (**A**) fractionation on an OH-40 column, active fraction G2 eluted at 20.90 min; (**B**) fractionation of G2 on a uno-Q column, the active fraction, Q1, not retained by the column; (**C**) fractionation of the Q1 fraction using the RP C18 column; and (**D**) fractionation of the P4 fraction with Aminiex HPX-87 h (the purified active fraction was named C4).

**Figure 3 cells-11-00869-f003:**
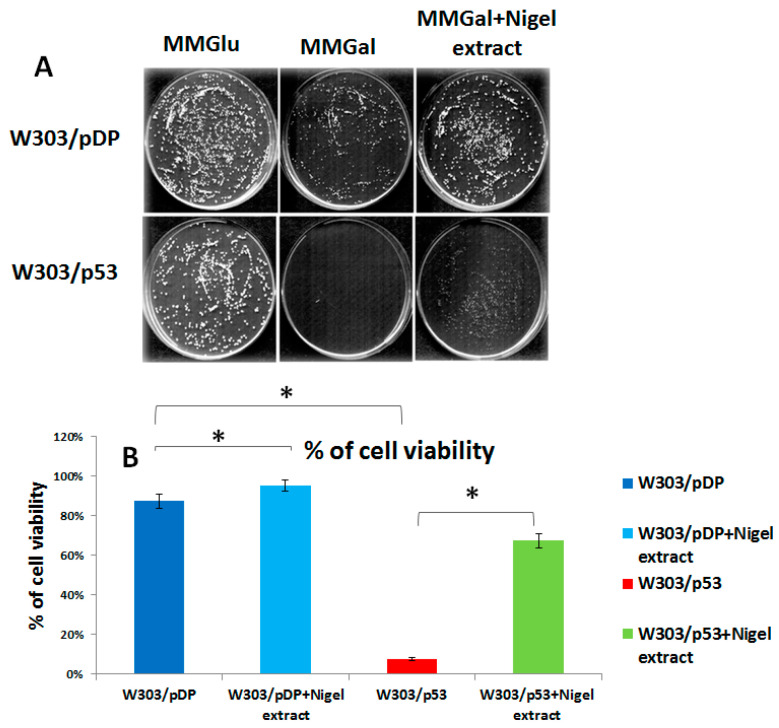
Quantitative analysis of the viability of clones W303/p53 and W303/pDP grown on MMGal (p53 induction) with and without aqueous *Nigella sativa L*. extract: (**A**) numbering of colonies of W303/pDP and W303/p53 plated on various media, (**B**) the percentage of cell viability was evaluated by counting the colonies. The percentage of the viable yeast cells recovered after p53 gene induction by galactose was calculated as follows: the number of viable cells in MMGal (+/– aqueous *Nigella sativa *L. extract/number of viable cells in glucose-containing solid MM) × 100. In the yeasts over-expressing p53, the cell viability was strongly and significantly decreased (Student’s *t*-test; *p* < 0.05); this decrease was strongly attenuated by the Nigella extract. * means that there is a significant differences.

**Figure 4 cells-11-00869-f004:**
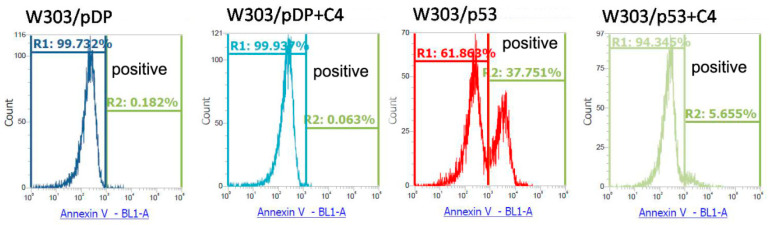
Quantification of apoptotic cells by flow cytometry. The apoptotic cells were quantified by flow cytometry by double staining with Annexin V-FITC and propidium iodide (PI). In this condition, Annexin V positive/PI negative cells are considered as apoptotic cells. The cytograms shown permit the quantification of the clones undergoing apoptosis by assessing the cells that are “Annexin V positive/PI negative”.

**Figure 5 cells-11-00869-f005:**
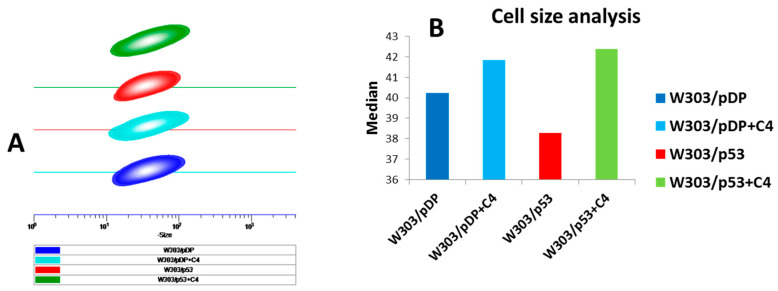
Flow cytometric analysis of the forward scatter channel. The forward scatter channel (FSC) presents the information concerning the cell size. (**A**) Yeast cells were pelleted, washed and re-suspended in PBS, and then analyzed by flow cytometry. (**B**) The histograms present the different median measurements of the FSC of W303/pDP and W303/p53 in the absence and presence of the C4 fraction.

**Figure 6 cells-11-00869-f006:**
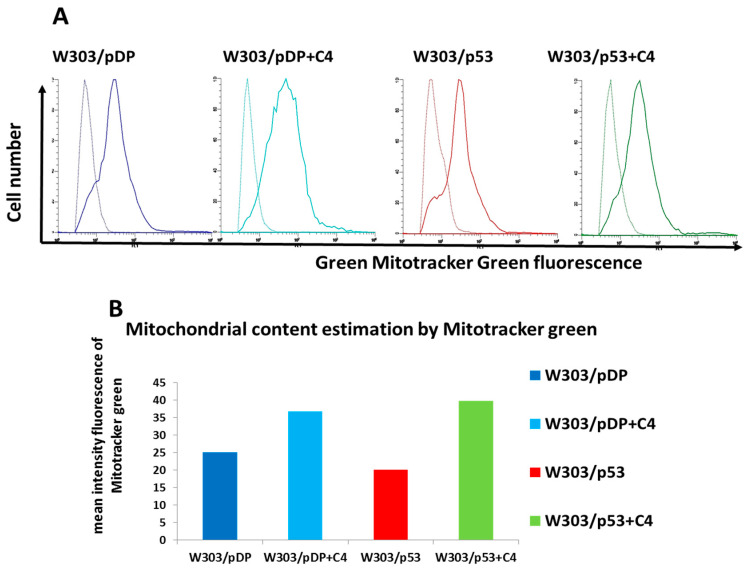
Effect of p53 over-expression and of the purified fraction from *Nigella sativa* L. extract on the mitochondria during apoptosis. The pDP and p53 transformed clones were grown in MMGal and in MMGal supplemented with the C4 fraction. The cells were pelleted, washed and suspended in PBS. The mitochondrial content in the cells was estimated by the specific mitochondria dye MitoTracker Green (FM). The cells were stained with 100 nM of MitoTracker Green and analyzed by flow cytometry. (**A**) The cytograms of the stained (continuous line) and unstained cells (dotted line). (**B**) The mitochondrial content is estimated by the difference of the mean intensity of fluorescence between the stained and unstained cells.

**Figure 7 cells-11-00869-f007:**
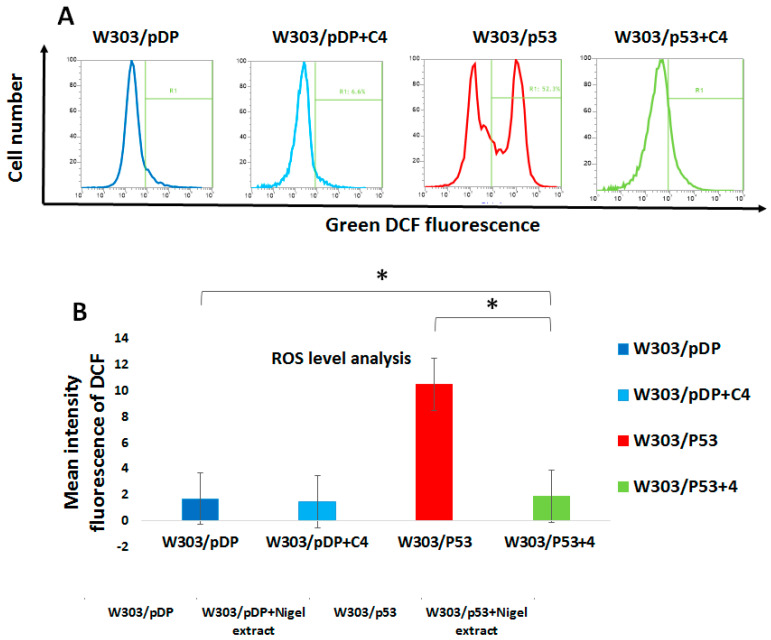
Intracellular ROS level measurement by flow cytometry in both W303/pDP and W303/p53, in the absence and presence of C4: W303/p53 and W303/pDP cells were grown in MMGal with and without the C4 fraction, and assessed by incubating the yeast cells with 10 mg/mL of H_2_DCF 2 h at 37 °C. (**A**) Cytograms of the stained (continuous line) and unstained cells (dotted line) and (**B**) the intracellular ROS is estimated by the difference in the fluorescence between the stained and unstained cells. ROS overproduction was significantly increased in W303/P53 yeasts (*p* < 0.05, Student’s *t*-test); this increase was strongly and significantly attenuated by the C4 fraction (the ROS level was similar to the control yeasts W303/pDP), and then in the control yeasts cultured in the presence of C4 fraction (W303/pDP + C4). * means that there is a significant differences.

**Figure 8 cells-11-00869-f008:**
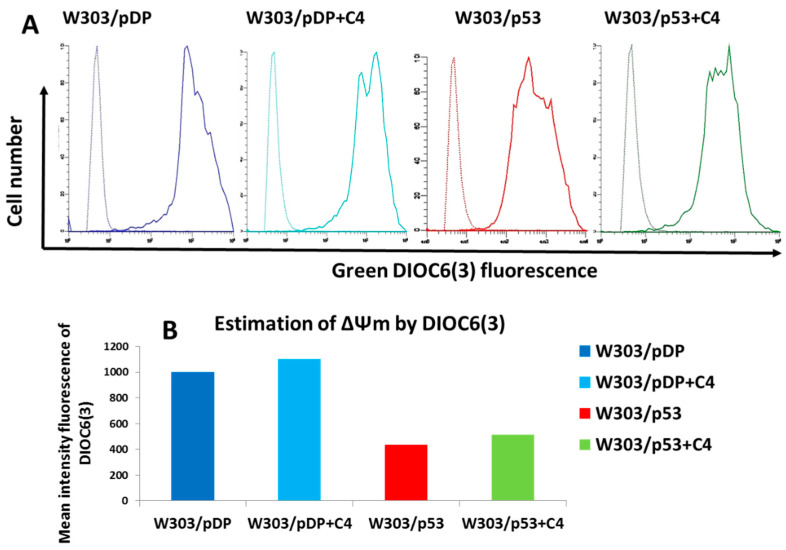
Evaluation of Δψm (mitochondrial membrane potential) by flow cytometry using the DiOC_6_(3) dye: W303/pDP and W303/p533 cells were cultured in MMGal, with or without added C4 fraction by cell staining with 40 nM of DiOC_6_(3). (**A**) Cytograms of stained (continuous line) and unstained (dotted line) cells, and (**B**) the mitochondrial membrane potential is estimated by the difference of the mean intensity fluorescence between the stained and unstained cells.

**Figure 9 cells-11-00869-f009:**
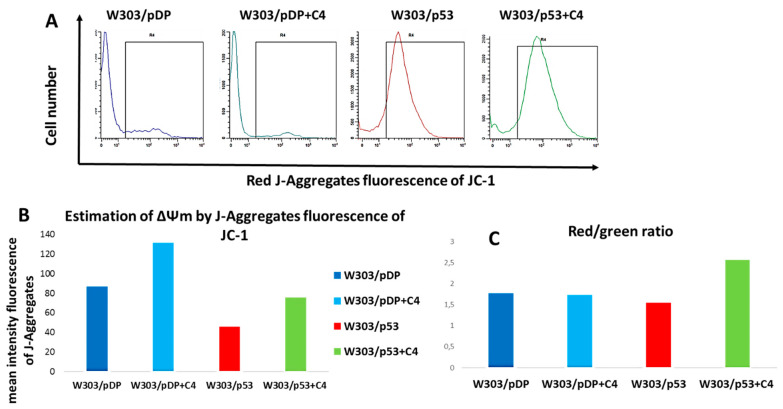
Mitochondrial membrane potential, Δψm, assessment by flow cytometry using the JC-1 dye: both W303/pDP and W303/p53 cells were cultured in MMGal, with or without added C4 fraction by staining with JC-1, (**A**) cytograms of the stained cells emitting red fluorescence of JC1-aggregates formed into mitochondria, (**B**) the mean intensity of JC1-aggregates in each strain, and (**C**) the red-to-green fluorescence ratio.

## Data Availability

All raw data are available from the Centre de Biotechnologie de Sfax, Sfax, Tunisia.

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
