# Peer review of "Whole and Purified Aqueous Extracts of Nigella sativa L. Seeds Attenuate Apoptosis and the Overproduction of Reactive Oxygen Species Triggered by p53 Over-Expression in the Yeast Saccharomyces cerevisiae"

_cells, 2022, doi:10.3390/cells11050869_

Round 1

Reviewer 1 Report

In the manuscript entipteld “Whole and purified aqueous extract of black cumin (Nigella sativa L) seeds attenuate apoptosis and reactive oxygen species overproduction triggered by p53 over-expression in Saccharomyces cerevisiae yeast” authors describe the effect of black cumin extract as well as of a specific fraction of the extract on yeast cells both expressing and not expressing mammalian p53. They show that either extract or a fraction C4 suppresses effect of p53 expression. 

The experiments are reasonably designed and appear to give reasonable data. The manuscript as a whole suffers from serious flaws that need to be addressed. 

1. English grammar and style must be improved throughout the manuscript. Manuscript needs to be carefully edited as many editing errors are in the whole manuscript (typos, punctuation, missing special characters, improperly formatted numbers (e.g. 10-5), format of references ……..)

2. I believe it is improper to call the situation the p53 is inducing in yeast the apoptosis. Although yeast certainly have specific forms of regulated cells death, they should not be called apoptosis, especially in the case of the ones artificially induced by expression of foreign proteins. (see Carmona-Gutierrez et al. Microb Cell. 2018 Jan 1; 5(1): 4–31.)

3. Discussion: “The addition of Nigella sativa L. molecule increased the survival of W303/p53, without affecting the p53 gene expression, suggested that the anti-apoptotic molecules acted on the consequence of the p53 effect, probably by scavenging/reducing the ROS in direct or indirect manner, and by leading to hyperpolarization of mitochondrial membrane. ”  

- Many effects of extract (e.g. on cell growth) can be observed even in the absence of p53 expression. It thus must not “act on the consequence of the p53”, it must rather affect cell physiology in a way that also affect the effects induced by p53.  (As you mention later in the text “Interestingly enough, the addition of Nigella sativa L. fraction engendered the same effects on the control strain (W303/pDP), namely the increase in cell size and cell content of mitochondria, the reduction of ROS and the increase of Δψm. This would mean that its action is independent of the presence of p53. ”) The discussion must reflect this.

4. Others:

Abstract:

“which do express and express human p53 under galactose inducer Gal 10 promoter control, respectively, were used. ” - ?? English !!!

Introduction:

“Apoptosis is defined as a programmed mechanism of cell death that is evolutionarily conserved.” - Not true. There are other ‘programmed mechanism of cell death that is evolutionarily conserved’ that are not apoptosis. (At least leave out word ‘defined’).

Materials:

“then serial dilutions from the cell culture were prepared ranging from 10-1 to 10-5. - What is “0-1 to 10-5” is it cells/ml ??

Results:

3.1 “Figure. 1B shows that in the presence of aqueous Nigel extracts, spread on galactose plates, that the growth of p53 recombined yeast cells was restored. ” 

- Should be fig 1A. Fig. 1B is growth in liquid culture.

3.4 “P53 over-expression reduced the survival on MMGal to 7% whereas it reduced the survival of 85% on W303/pDP yeast cells (Figure. 3). In presence of aqueous Nigella sativa L. extracts Fig. 2), plated on MMGal, cell viability of the recombinant yeast W303/p53 increased to 65.5% and that of W303/pDP was also improved from 88% to 95% (Figure. 3). ”

- Why 88% to 95%? (Shouldn’t it be 85% to 95%?)

3.10 “Furthermore red to green ratio of JC-1 is considered for the direct assessment of mitochondria polarization and was indeed lower in W303p53 strain; however, it is high in W303pDP strain and in W303p53 in presence of C4 fraction (figure. 9C). This result confirms that mitochondrial membrane potential is lower in W303/p53 and that Nigella sativa L. fraction prevents such modification.” 

-Although it appears to me that what you refer to as transmembrane potential (red/green ratio with JC-1) is elevated in p53+C4 cells, I wold not say it is lower in p53 cells 

Author Response

Reviewer 1

  • In the manuscript entitled “Whole and purified aqueous extract of black cumin (Nigella sativa L) seeds attenuate apoptosis and reactive oxygen species overproduction triggered by p53 over-expression in Saccharomyces cerevisiae yeast” authors describe the effect of black cumin extract as well as of a specific fraction of the extract on yeast cells both expressing and not expressing mammalian p53. They show that either extract or a fraction C4 suppresses effect of p53 expression. 

The experiments are reasonably designed and appear to give reasonable data. The manuscript as a whole suffers from serious flaws that need to be addressed. 

  1. English grammar and style must be improved throughout the manuscript. Manuscript needs to be carefully edited as many editing errors are in the whole manuscript (typos, punctuation, missing special characters, improperly formatted numbers (e.g. 10-5), format of references ……..)
  • We checked the entire manuscript and we corrected what has been asked
  1. I believe it is improper to call the situation the p53 is inducing in yeast the apoptosis. Although yeast certainly have specific forms of regulated cells death, they should not be called apoptosis, especially in the case of the ones artificially induced by expression of foreign proteins. (see Carmona-Gutierrez et al. Microb Cell. 2018 Jan 1; 5(1): 4–31.)
  • In fact, to determine whether a yeast cell has been engaged in an apoptotic pathway, some methods have been used such as Annexin V/PI of spheroblasts. This was previously proven by our group that the P53 over expression leads to the externalization of PS of the plasma membrane in transformed yeasts when cultivated in the minimal media, in addition of ROS overproduction and DNA strand cleavage (Yacoubi Hadj-Amor et al. Human P53 induces cell death and downregulates thioredoxin expression in Saccharomyces cerevisiae (DOI:10.1111/j.1567-1364.2008.00445.x). It is admitted that in mammalian cells deprived of nutrients such as serum, the p53 activity and the resulting signaling pathways are strongly induced (Prives C & Hall PA (1999) The p53 pathway. J Pathol 187: 112–126.), but still always P53 under various stresses (DNA damage, irradiation, oxidative stress, etc..) induces apoptosis. Similarly, the human wtp53 is active in cerevisiae and have cytoplasmic, nuclear and mitochondrial localization and triggers the release of cytochrome c, only when it is full-length, i.e, not deleted from its nuclear localization signal (Abdelmoula et al, Cellular localization of human p53 expressed in the yeast Saccharomyces cerevisiae: effect of NLSI deletion). Thus, the corresponding cell death is more closed to apoptosis than necrosis or autophagy. Nevertheless, we agree to refer to p53-induced cell death in yeast as 'cell death' or 'regulated cell death' (RCD) as in the cited review, instead of 'apoptosis'. We prefer the term RCD to the terms ATCD or ADCD. In the actual work, we carefully targeted physiological and phenotypical modifications that are generally defined as apoptotic characteristics.
  1. Discussion: “The addition of Nigella sativa L. molecule increased the survival of W303/p53, without affecting the p53 gene expression, suggested that the anti-apoptotic molecules acted on the consequence of the p53 effect, probably by scavenging/reducing the ROS in direct or indirect manner, and by leading to hyperpolarization of mitochondrial membrane. ” 

- Many effects of extract (e.g. on cell growth) can be observed even in the absence of p53 expression. It thus must not “act on the consequence of the p53”, it must rather affect cell physiology in a way that also affect the effects induced by p53.  (As you mention later in the text “Interestingly enough, the addition of Nigella sativa L. fraction engendered the same effects on the control strain (W303/pDP), namely the increase in cell size and cell content of mitochondria, the reduction of ROS and the increase of Δψm. This would mean that its action is independent of the presence of p53. ”) The discussion must reflect this.

We replaced « acted on the consequence of the p53 effect >> by « must rather affect cell physiology in a way that also affect the effects induced by p53, »

  1. Others:

Abstract:

“which do express and express human p53 under galactose inducer Gal 10 promoter control, respectively, were used. ” - ?? English !!!

  • We corrected the phrase as follows:

To this end, the Saccharomyces cerevisiae W303-1B strain, transformed by the pDP vector (W303pDP) and the recombinant strain (W303p53) transformed by pDP-p53 plasmid, expressing p53 under the Gal10 promotor inducible by galactose,  were used

  • Introduction:

“Apoptosis is defined as a programmed mechanism of cell death that is evolutionarily conserved.” - Not true. There are other ‘programmed mechanism of cell death that is evolutionarily conserved’ that are not apoptosis. (At least leave out word ‘defined’).

  • We agree and we have consequently modified the sentence: “Apoptosis can be described as an evolutionarily conserved cell death mechanism “
  • Materials:

“then serial dilutions from the cell culture were prepared ranging from 10-1 to 10-5. - What is “0-1 to 10-5” is it cells/ml ??

  • It was modified as follows: “then a ten-fold serial dilutions from the cell culture were prepared “
  • Results:

3.1 “Figure. 1B shows that in the presence of aqueous Nigel extracts, spread on galactose plates, that the growth of p53 recombinant yeast cells was restored. ” 

- Should be fig 1A. Fig. 1B is growth in liquid culture.

  • It’s true and it’s corrected
  • 4 “P53 over-expression reduced the survival on MMGal to 7% whereas it reduced the survival of 85% on W303/pDP yeast cells (Figure. 3). In presence of aqueous Nigella sativa L. extracts Fig. 2), plated on MMGal, cell viability of the recombinant yeast W303/p53 increased to 65.5% and that of W303/pDP was also improved from 88% to 95% (Figure. 3). ”

- Why 88% to 95%? (Shouldn’t it be 85% to 95%?)

  • We corrected the sentence: “viability was assessed by the percentage of “Colony Forming Units”. This percentage was of 85% in W303/pDP and 7% in W303/p53. In presence of aqueous Nigella sativa extracts (Figure. 3), plated on MMGal, cell viability of the recombinant yeast W303/p53 increased to 65.5% and that of W303/pDP was also improved to 95%” .

10 “Furthermore red to green ratio of JC-1 is considered for the direct assessment of mitochondria polarization and was indeed lower in W303p53 strain; however, it is high in W303pDP strain and in W303p53 in presence of C4 fraction (figure. 9C). This result confirms that mitochondrial membrane potential is lower in W303/p53 and that Nigella sativa L. fraction prevents such modification.” 

-Although it appears to me that what you refer to as transmembrane potential (red/green ratio with JC-1) is elevated in p53+C4 cells, I would not say it is lower in p53 cells 

  • For JC-1, mitochondrial membrane potential is characterized by the red fluorescence of aggregates formed in mitochondria with high membrane potential (living cells). If this potential is low, there are no aggregates but only monomers inside and outside the mitochondria. The green fluorescence of the monomers is always present in greater percentage because their quantity in the cytoplasm is high, therefore the difference between the red/green ratios is not always great. The red/green fluorescence ratio can be considered as an assessment of the state of the mitochondria polarization. Nevertheless, J-aggregates that form at precise locations, still the ideal tool for investigating changes of mitochondrial membrane potential âˆ†Ψ In cells with low ∆Ψm, JC-1 exists in its monomeric form, showing only green fluorescence. The cells with high ∆Ψm form J-aggregates, inside the mitochondria, thus show red fluorescence. In cells with low ∆Ψm, JC-1 exists in its monomeric form, and showing only green fluorescence located in cytoplasm and mitochondrial matrix in depolarized mitochondria.

Reviewer 2 Report

Authors submitted paper entitled "Whole and purified aqueous extract of black cumin (Nigella sativa L) seeds attenuate apoptosis and reactive oxygen species overproduction trig-gered by p53 over-expression in Saccharomyces cerevisiae yeast". In general is well dane and proposed experimental setup. Despite many effort authors made few mistake, which below I try point by point show:

  1. Proposed by Authors survival test is not show direct survive. It show only clonogenity, Authors indicate that "Colony forming units (cfu) were quantified." Therefore please suply this assay for e.g.iodium propidium dye. On figure 3 is needed change this calculate. For my opinion Authors will add additional figure present real viability. Present Figure 3 should change description as a clonogenity test/ability to clon formation. Please mark statistical significantly on figure.
  2. Figure 1 A please add density in 'serial dillutions'. 
  3. W303 is different and specific background. Therefore please provide evidence to figure 5, that this size is not associated or associated with vacuole fusion. For my opinion it will be strongly corelated with vacuole size.fusion. Fig 5B - please delate figure description - it is mention in figure legend. Lack statistical analysis. Y axis - median (unit)
  4. Figure 6, 8, 9 - lack statistical analysis. 
  5. Figure 7 - please mark statistical significance on figure.
  6. In general unify all figures - delate figure description e.f. ROS level analysis etc. 

Round 2

Reviewer 1 Report

Revised version of manuscript entitled “Whole and purified aqueous extract of Nigella sativa L seeds attenuate apoptosis and overproduction of reactive oxygen species triggered by p53 over-expression in the yeast Saccharomyces cerevisiae.” is significantly improved as compared to the original manuscript. There are, however, still several issues that needs to be addressed.

  1. Abstract should be rewritten. There no need to be specific concerning names of plasmids, strains, etc. It should be rather written in descriptive language that should be easy to read even to those outside the specific field.

  1. Although the language was improved and flow of text is now good, I still feel that manuscript would profit from revision by a native speaker and from a careful edition of the text. Here I list some of my suggestions, but revision should not be limited to those:

Line 76

“Apoptosis can be described as an evolutionarily conserved cell death mechanism.” should be “Apoptosis is as an evolutionarily conserved cell death mechanism. “

Line 94

“MOMP (mitochondrial outer membrane)” should be “MOMP (mitochondrial outer membrane permeabilization)”

Line 97-99

“Once outside the mitochondria, cytochrome-c binds to APAF-1 (Apoptotic Protease Activating Factor-1) acting as the adapter molecule responsible of caspases activation. ” I believe cytochorme c is not an adaptor. It binds to Apaf-1 , which induces change in Apaf-1 that enables oligomerization of Apaf-1. 

Line 310

“alpha” character is missing in genotype of W303-1B

Line 322

“The growth kinetics was performed on solid ..” should be “The growth kinetics was measured on solid ..”

Line 1545-6

“Nevertheless, J-aggregates that form at precise locations, still the ideal tool for investigating changes of mitochondrial membrane potential ∆Ψm. “ - incomplete sentence

Author Response

ANSWER TO REVIEWER 1

Revised version of manuscript entitled “Whole and purified aqueous extract of Nigella sativa L seeds attenuate apoptosis and overproduction of reactive oxygen species triggered by p53 over-expression in the yeast Saccharomyces cerevisiae.” is significantly improved as compared to the original manuscript. There are, however, still several issues that needs to be addressed.

  1. Abstract should be rewritten. There no need to be specific concerning names of plasmids, strains, etc. It should be rather written in descriptive language that should be easy to read even to those outside the specific field.

 The abstract has been rewritten

  1. Although the language was improved and flow of text is now good, I still feel that manuscript would profit from revision by a native speaker and from a careful edition of the text. Here I list some of my suggestions, but revision should not be limited to those:

We have reviewed the text and we did the required modifications

Line 76

“Apoptosis can be described as an evolutionarily conserved cell death mechanism.” should be “Apoptosis is as an evolutionarily conserved cell death mechanism. “

 It is done

Line 94

“MOMP (mitochondrial outer membrane)” should be “MOMP (mitochondrial outer membrane permeabilization)”

 It is done

Line 97-99

“Once outside the mitochondria, cytochrome-c binds to APAF-1 (Apoptotic Protease Activating Factor-1) acting as the adapter molecule responsible of caspases activation. ” I believe cytochorme c is not an adaptor. It binds to Apaf-1 , which induces change in Apaf-1 that enables oligomerization of Apaf-1. 

We agree, we have modified the sentence.

Line 310

“alpha” character is missing in genotype of W303-1B

 It is corrected

Line 322

“The growth kinetics was performed on solid ..” should be “The growth kinetics was measured on solid ..”

It is corrected

Line 1545-6

“Nevertheless, J-aggregates that form at precise locations, still the ideal tool for investigating changes of mitochondrial membrane potential ∆Ψm. “ - incomplete sentence

It is corrected

Reviewer 2 Report

Well done!

Author Response

ANSWER TO REVIEWER 2

No modifications were required.
